# Otolith geochemistry reflects life histories of Pacific bluefin tuna

John A. Mohan[1,2]*, Heidi Dewar[3], Owyn E. Snodgrass[3], Nathan R. Miller[4], Yosuke Tanaka[5], Seiji Ohshimo[5,6], Jay R. Rooker[1], Malcom Francis[7], R. J. David Wells[1]

1 Department of Marine Biology, Texas A&M University at Galveston, Galveston, Texas, United States of America, 2 School of Marine and Environmental Programs, University of New England, Biddeford, Maine, United States of America, 3 Southwest Fisheries Science Center, National Marine Fisheries Service, La Jolla, California, United States of America, 4 Jackson School of Geosciences, The University of Texas at Austin, Austin, Texas, United States of America, 5 Highly Migratory Resources Division, Fisheries Stock Assessment Center, Fisheries Resources Institute, Fisheries Research and Education Agency, Shizuoka, Japan, 6 Pelagic Fish Resources Division, Fisheries Stock Assessment Center, Fisheries Resources Institute, Fisheries Research and Education Agency, Nagasaki, Japan, 7 National Institute of Water and Atmospheric Research (NIWA),Wellington, New Zealand

* jmohan@une.edu

**Data Availability Statement:** All relevant data are within the paper and its Supporting Information files.

## Abstract

Understanding biological and environmental factors that influence movement behaviors and population connectivity of highly migratory fishes is essential for cooperative international management and conservation of exploited populations, like bluefin tuna. Pacific bluefin tuna *Thunnus orientalis* (PBT) spawn in the western Pacific Ocean and then juveniles disperse to foraging grounds across the North Pacific. Several techniques have been used to characterize the distribution and movement of PBT, but few methods can provide complete records across ontogeny from larvae to adult in individual fish. Here, otolith biominerals of large PBT collected from the western, eastern, and south Pacific Ocean, were analyzed for a suite of trace elements across calcified/proteinaceous growth zones to investigate patterns across ontogeny. Three element:Ca ratios, Li:Ca, Mg:Ca, and Mn:Ca displayed enrichment in the otolith core, then decreased to low stable levels after age 1–2 years. Thermal and metabolic physiologies, common diets, or ambient water chemistry likely influenced otolith crystallization, protein content, and elemental incorporation in early life. Although similar patterns were also exhibited for otolith Sr:Ca, Ba:Ca and Zn:Ca in the first year, variability in these elements differed significantly after age-2 and in the otolith edges by capture region, suggesting ocean-specific environmental factors or growth-related physiologies affected otolith mineralization across ontogeny.

## Introduction

Characterizing the complete life histories of highly migratory species is challenging due to their long-distance movements through remote habitats and difficulties in observing them. Knowledge of migration routes and population connectivity is important for effective

**Funding:** This work was funded by a grant from the National Oceanic and Atmospheric Administration, U.S. Department of Commerce, Saltonstall-Kennedy research program (grant # NA17NMF4270224) to RJDW, JAM, and JRR. The statements, findings, conclusions, and recommendations are those of the author(s) and do not necessarily reflect the views of the National Oceanic and Atmospheric Administration or the Department of Commerce.

**Competing interests:** The authors have declared that no competing interests exist.

(spatially explicit) management and conservation [1], especially for species that undertake transoceanic movements and cross international boundaries, where they are susceptible to fishery exploitation by multiple nations [2, 3]. A prime example is the Pacific bluefin tuna (*Thunnus orientalis*). Pacific bluefin tuna (PBT) exhibit a suite of adaptations enabling fast swimming speeds, and expansive migrations that connect disparate oceanic ecosystems across hemispheres.

Methods such as electronic and conventional tagging and analysis of catch data have enhanced our knowledge of life-stage specific behaviors of PBT [4, 5]. Documented spawning of PBT occurs only in the western Pacific Ocean (WPO) [4] around the Philippines and East China Sea in April-June and in the Sea of Japan in July-August [6, 7]. PBT spawning in the East China Sea are 8+ years old, while those spawning in the Sea of Japan are typically age 3–6 years [7, 8]. Larval PBT spawned in the East China Sea and near the Philippines are then transported northward by the Kuroshio Current to utilize coastal areas as nurseries along the southern coast of Japan [9, 10] and in the Sea of Japan [11]. Fish spawned in the Sea of Japan may remain in local waters seeking preferred temperatures ranging from 23 to 26˚C [11]. Juvenile PBT (age 0–2) then seek favorable thermal habitats with abundant prey in the vicinity of the Kuroshio Current [12]. Some one to two year old PBT undergo transoceanic migrations to the eastern Pacific Ocean (EPO), with migration journeys ranging from 1.2 to 5.5 months and departure timings dependent on nursery foraging areas [6, 12]. PBT remain in the EPO for several years (ranging from 3 to 9) before returning to the WPO [13].

In addition to the EPO, historical catch records indicate some portion of the population also migrates across the equator into tropical south Pacific waters. For example, large PBT have been caught in long line and recreational fisheries of New Zealand [14, 15]. While previous tagging studies provide useful information on movements of PBT to the EPO and the SPO, these techniques are limited to fish behavior post-tagging, and do not provide birth-to-capture life histories. In contrast, natural tags such as the molecular and elemental compositions of fish tissues and hard parts, can offer unique opportunities to reconstruct more complete life histories.

Natural tags have been applied to investigate migration dynamics in PBT. Some studies examine the isotopic composition of muscle tissue including nitrogen stable isotopes [16, 17] and Fukushima-derived radionuclides [18]. As a metabolically active tissue, isotopes in muscle provide a time-window that is equal to the turnover rate [19]. For PBT, this time window is approximately a year, which limits the scale of question that can be addressed.

In comparison, calcified structures such as fish otoliths (ear stones) form continuously by radial growth and are metabolically inert once formed, thus otolith core-to-edge chemical compositions span entire life histories of specimens. Otoliths grow by accretion of calcium carbonate crystals on a protein matrix, within endolymph fluid that is influenced by blood chemistry [20–22]. Daily physiological processes can affect aspects of blood chemistry (e.g. pH, bicarbonate, protein content) [20, 23]. Environmental conditions such as temperature, salinity, and dissolved oxygen [24] can also influence blood chemistry, and consequently some elements incorporated within the calcium carbonate structure are useful environmental proxies. There is high uncertainty on the relative influence of intrinsic and extrinsic processes that affect element incorporation into fish otolith and responses are species specific [24]. Few studies have yet examined the influence of endothermic physiology on crystal formation biochemistry in pelagic fish species, due to the difficulty of conducting controlled experiments [25]. Bluefin tuna are regional endotherms that can elevate the temperature of their eyes and brain [26], therefore thermal physiology may influence otolith biomineral element incorporation and confound interpretation of fish movement.

To the extent that PBT move through geochemically distinct water masses and physiological influences on biomineralization can be accounted for, PBT otolith geochemical records could allow for reconstruction of migratory movements and identification of migratory contingents. Previous studies utilized otolith chemistry to characterize signatures of young-of-the-year PBT collected on spawning grounds [27, 28], to determine natal origins of PBT that migrated to the EPO [29], to document discrete profiles of juveniles collected in the EPO [30], and assess the timing of juvenile emigration from the WPO [31]. However, studies documenting continuous life histories for large PBF otoliths collected over wide areas of the north and south Pacific are needed to better understand influences of physiology and water mass environmental conditions on otolith chemical time series.

Here, otoliths of large PBT collected from the WPO, EPO and SPO were analyzed using laser-ablation inductively coupled plasma mass spectrometry (LA-ICP-MS) to document chemical chronologies from juvenile to adult life in sequentially deposited growth bands. By comparing complete elemental profiles of PBT sourced from different ocean basins, this study explores if 1) certain elements reflect ocean basin-scale migratory movements or regional residence and 2) how physiology, the environment and *in situ* water chemistries influence the element patterns across ontogeny in PBT otoliths.

## Materials and methods

### Otolith collection

PBT otoliths (n = 25) were collected from fishery-dependent sources in three regions of the Pacific Ocean: the eastern Pacific Ocean (EPO, n = 10), the western Pacific Ocean (WPO, n = 10) and south Pacific Ocean (SPO, n = 5) (Fig 1, Table 1).

As some samples were collected opportunistically, it was not always possible to obtain full metadata for each tuna, as sometimes only the fish head was available. Region of collection (EPO, WPO, SPO) was used to compare elemental patterns. Where possible, the fork length (FL cm), weight (kg), sex and capture location were recorded. For EPO samples where only heads were available, the operculum length (OL cm) was used to estimate FL using the equation: FL = OL*3.802–13.794; $r^2$ = 0.94 [33]. Following collection, dried, tissue-free otoliths were stored in labeled plastic vials. Whole sagittal otoliths were embedded in a clear epoxy resin EpoFix (Stuers) so that the distal lobe could be used to identify the location of the core. The resin was spiked with 30 ppm indium ($^{115}$In) during mixing to serve as an internal elemental marker of the epoxy. Embedded otoliths were sectioned using a low-speed diamond blade saw to obtain a ~1 mm thick central section from the transverse plane (perpendicular to the longest otolith axis, Fig 2).

Central otolith sections were mounted on a petrographic slide using thermoplastic (Crystalbond™) adhesive, with the distal lobe facing up, then surface polished using 600–1200 grit silicone-carbide paper (Buehler) and ultrapure water until the distal lobe became transparent and the sulcus groove formed a sharp narrow 'V', indicating the core was reached (Fig 2). Final polished sections were remounted on new petrographic slides, such that each slide contained several closely spaced otoliths arranged in a random sequence. The slides were scanned at high-resolution to assist with placement of laser transects.

### Elemental measurements and data analysis

Elemental concentrations were measured using a New Wave 193 nm laser coupled to an 7500ce Agilent inductively coupled plasma mass spectrometer at the University of Texas at Austin. All samples and standards were loaded into a large format cell with fast washout times (< 1 s). All laser scans began in the otolith core and moved outwards along the longest growth

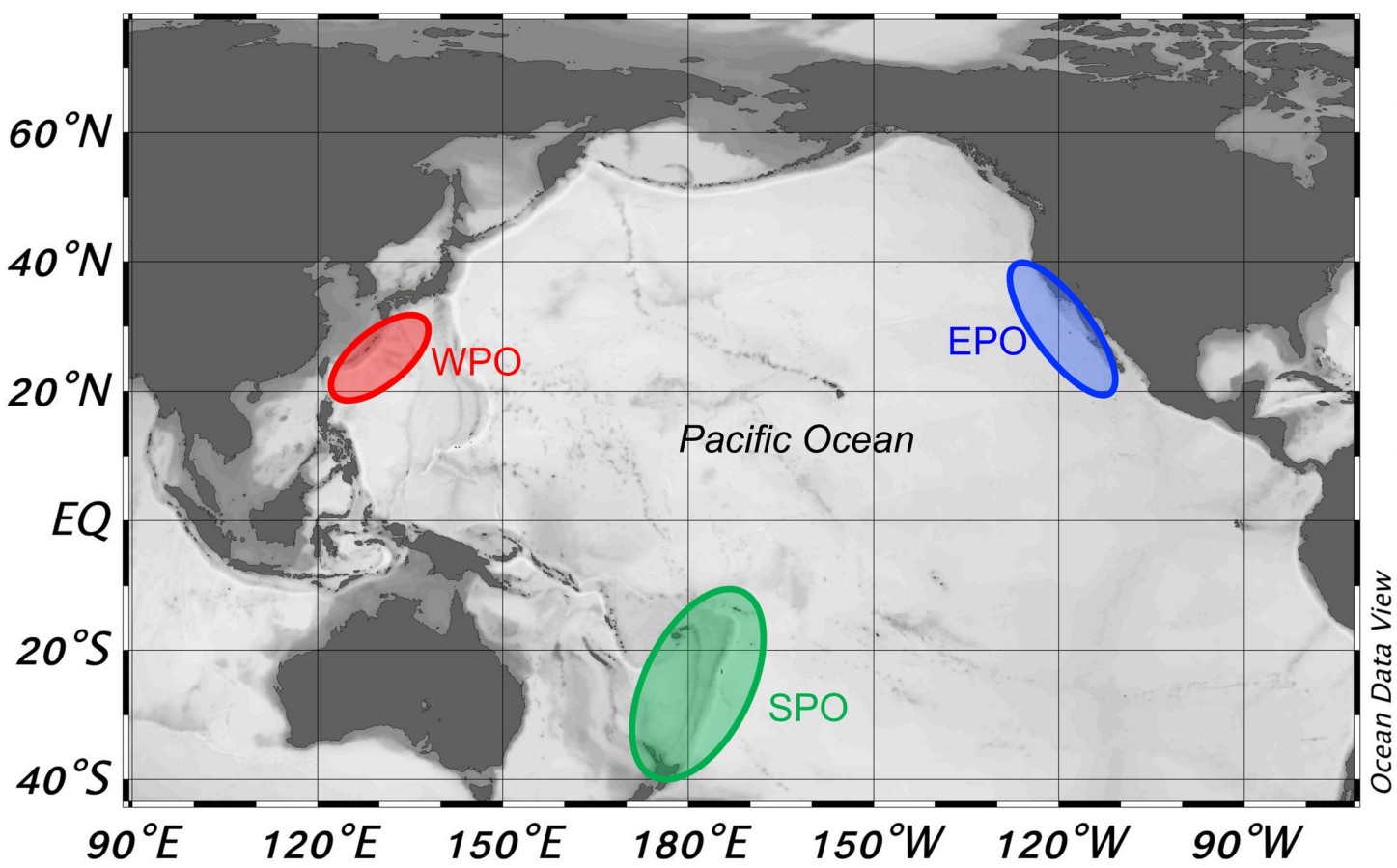

**Fig 1. Geographical map of opportunistic collected PBT otoliths samples from the eastern Pacific (EPO), western Pacific (WPO) and south Pacific (SPO); different colored ellipses represent approximate collection regions.**

axis (Fig 2). Test scans revealed optimal ion counts using gas flows of 850 mL min$^{-1}$ for Ar and 800 mL min$^{-1}$ for He. Otolith sections and standards were pre-ablated to remove any surface contamination using a 75μm spot, 50μm s$^{-1}$ scan rate, 20 Hz repetition rate and 40% power. Data acquisition parameters were 35% power, 20 Hz with a 50 μm spot moving at 5μm s$^{-1}$. Laser fluence during analysis averaged 2.12 ± 0.02 J/cm$^2$. The quadrupole time-resolved method measured 13 masses using integration times of 10 ms ($^{24}$Mg, $^{43\text{-}44}$Ca, $^{88}$Sr, $^{115}$In), 20 ms ($^{25}$Mg, $^{55}$Mn), and 50 ms ($^{7}$Li, $^{59}$Co, $^{63}$Cu, $^{66}$Zn, $^{137\text{-}138}$Ba). Time-resolved intensities were converted to concentration (ppm) equivalents using Iolite software (Univ. Melbourne, [34]), with $^{43}$Ca as the internal standard and a Ca index value of 38.3 weight %. Baselines were determined from 30-s gas blank intervals measured while the laser was off, and all masses were scanned by the quadrupole. USGS MACS-3 was used as the primary reference standard and accuracy and precision were proxied from replicates of NIST 612 analyzed as an unknown. NIST 612 analyte recoveries were typically within 2% of GeoREM preferred values (http://georem.mpch-mainz.gwdg.de).

Concentration data (ppm) were converted to molar ratios to facilitate comparisons with previous studies. To remove high frequency noise, time-series were smoothed by sequential application of 7-point moving median and 7-point moving average filters. In order to remove edge effects associated with intersection of the laser with epoxy resin, otolith edges were defined at crossover points where $^{43}$Ca and $^{115}$In counts-per-second (CPS) were < 200,000

**Table 1. Pacific bluefin tuna (PBT) metadata including date, ocean region of collection region, local region of collection; sex, gilled and gutted (GG) weight and fork length (FL) reported where available.** Estimated ages were derived from length-age relationships reported in [32]. Laser distance measured as in Fig 2B.

| PBT_ID | Date PBT collected | Ocean PBT collected | Local region PBT collected | Sex | FL (cm) | Estimated age (yr) | GG weight (kg) | Laser distance (μm) |
|---|---|---|---|---|---|---|---|---|
| EP01 | 9/20/17 | East Pacific | San Deigo, Califronia USA | M | 197 | 9 | 156 | 3086 |
| EP02 | 7/29/17 | East Pacific | San Deigo, Califronia USA | | 182 | 7+ | 167 | 3165 |
| EP03 | 7/20/17 | East Pacific | San Deigo, Califronia USA | | 179 | 7 | | 3258 |
| EP04 | 8/8/16 | East Pacific | San Deigo, Califronia USA | | 180 | 7 | | 2818 |
| EP05 | 8/8/16 | East Pacific | San Deigo, Califronia USA | | 180 | 7 | | 3310 |
| EP06 | 10/30/17 | East Pacific | San Deigo, Califronia USA | | 178 | 7 | 103 | 3284 |
| EP07 | 7/29/16 | East Pacific | San Deigo, Califronia USA | F | 181 | 7+ | | 2929 |
| EP08 | 7/19/17 | East Pacific | San Deigo, Califronia USA | | 173 | 6+ | | 3127 |
| EP09 | 7/7/17 | East Pacific | San Deigo, Califronia USA | | 173 | 6+ | | 2957 |
| EP10 | 8/20/16 | East Pacific | San Deigo, Califronia USA | | 174 | 7 | | 2726 |
| WP01 | 5/12/17 | West Pacific | Nansei Islands | F | 225 | 13 | 219 | 3342 |
| WP02 | 5/5/17 | West Pacific | Nansei Islands | F | 222 | 12 | 205 | 3355 |
| WP03 | 5/8/17 | West Pacific | Nansei Islands | F | 210 | 10 | 167 | 3345 |
| WP04 | 5/7/17 | West Pacific | Nansei Islands | F | 208 | 10 | 160 | 2903 |
| WP05 | 5/4/17 | West Pacific | Nansei Islands | F | 206 | 10 | | 3083 |
| WP06 | 5/11/17 | West Pacific | Nansei Islands | M | 222 | 12 | 214 | 3425 |
| WP07 | 5/10/17 | West Pacific | Nansei Islands | M | 199 | 9 | 156 | 3145 |
| WP08 | 5/12/17 | West Pacific | Nansei Islands | F | 223 | 13 | 241 | 3748 |
| WP09 | 5/12/17 | West Pacific | Nansei Islands | | 203 | 9 | 154 | 3230 |
| WP10 | 5/10/17 | West Pacific | Nansei Islands | F | 212 | 10+ | 156 | 3439 |
| NZ01 | 8/22/07 | South Pacific | New Zealand | F | 245 | 22 | 180 | 4187 |
| NZ02 | 5/6/13 | South Pacific | New Zealand | F | 168 | 6 | 100 | 3212 |
| SP01 | 3/31/18 | South Pacific | Cook Islands | M | 265 | 26+ | | 4052 |
| SP02 | 8/15/17 | South Pacific | Cook Islands | M | 228 | 14 | 255 | 3599 |
| SP03 | 8/23/18 | South Pacific | Cook Islands | F | 253 | 26+ | | 4135 |

and > 1000 CPS, respectively. Using these criteria, there were some instances when Mg:Ca and Li:Ca increased shortly before the edge, which was likely due to thermoplastic Crystalbond™ cement penetrating the intersection of the otolith-epoxy edge. One otolith (NZ02) clearly had translucent Crystalbond™ covering the surface that was confirmed after analysis, which affects the Mg:Ca values. Therefore, Mg:Ca was not plotted for this specimen (see S1 Fig). Approximate core-to-edge distances of mean annuli distances along the laser path (see Fig 2 in [35]) were measured in Image J and superimposed on elemental time-series plots to discern how elemental patterns generally relate to age. The mean annuli distances were 1,223 μm in year one, 1,577 μm in year two, 2,054 μm for year 3+, as measured from the otolith core and along the laser path.

To document regional spatial variation in elemental signatures at times of capture, the molar element:Ca ratios at the edge of the otoliths were compared among EPO, WPO and SPO collection regions. For each otolith, the final 100 μm of the otolith edge was averaged to represent the recent otolith material accreted in the region of collection, which represents different time frames for individual fish. The otolith edge data was then inspected for outliers using a Grubbs test, with identified outliers removed from calculated averages and subsequent statistical analysis. Normality was assessed using a Kologorov-Smirnov test and only Mg:Ca was not normal distributed, and thus a log transform was used to meet normality assumption for Mg:Ca. A Brown-Forsythe test confirmed that the standard deviations were not significantly different among regions and thus the parametric ANOVA was appropriate, using

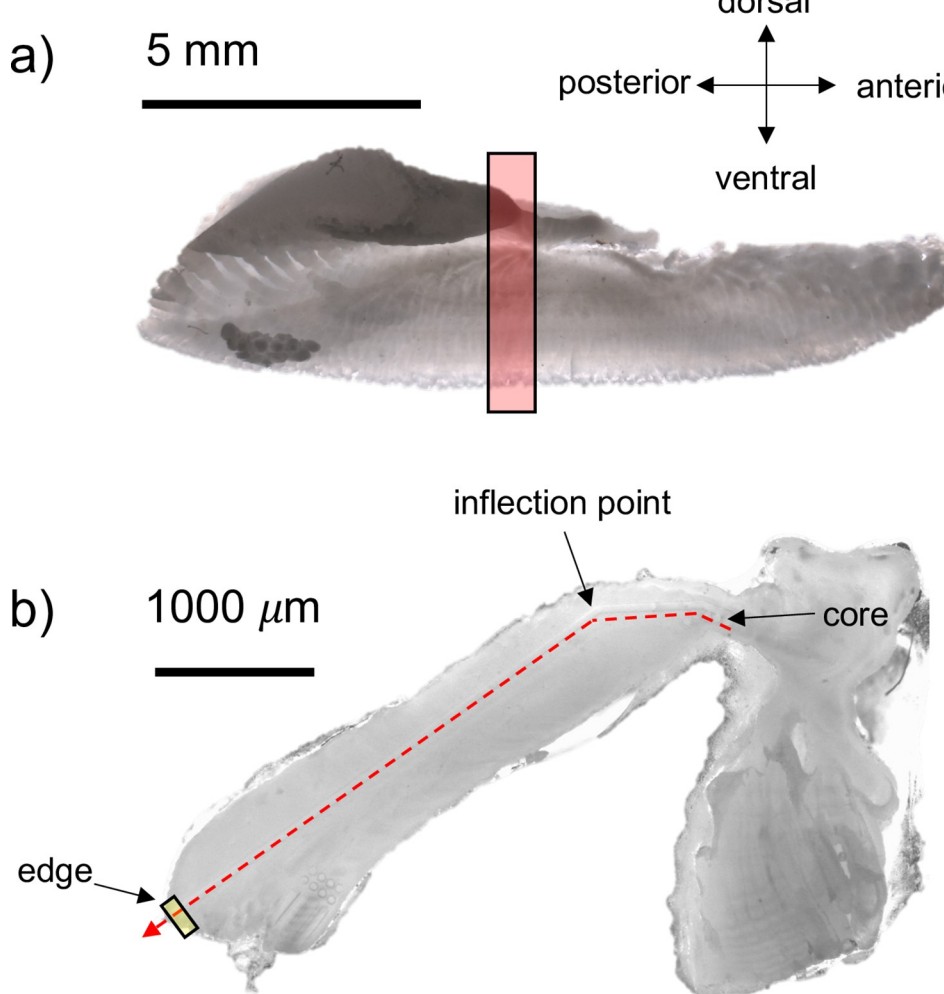

**Fig 2. Schematic of Pacific bluefin tuna otolith preparation for elemental analysis.** Otoliths of PBT were sectioned in the transverse plane (a, red shaded box) and the ~1 mm section was polished to expose the core; a laser scan (b, red dashed line) starting from the early life core region was scanned across the otolith growth bands encompassing the complete life history, with the laser path turning at the inflection point; the final 100 μm of data (small yellow box) was averaged to represent the recent otolith material deposited in the region of collection.

PRISM 8 statistical program. To test if elemental differences were present among each region, a one-way analysis of variance (ANOVA) test was performed for followed by a Holm-Sidak multiple comparison test. For element:Ca ratios that were significantly different among regions, quadratic discriminant function analysis (QDFA) was used to determine whether PBT from each region could be chemically identified using a jackknifed classification matrix using SYSTAT 13. Due to low sample size from the SPO (N = 5), the number of elements used in the QDFA had to be limited to three. The first and second canonical scores were then plotted to visualize separations among regions.

## Results

Study otoliths were collected from 25 PBT: 10 from EPO, 10 from WPO and 5 from SPO (Fig 1, Table 1). EPO and WPO collections were from 2016 and 2017, whereas SPO collection spanned from 2007 to 2018. Estimated specimen ages (based on length from Shimose et al.

2009) ranged from 6 to 26 year with most young fish coming from the EPO and one age 6 specimen from the SPO (Table 1).

## Mean element:Ca profiles

Element:Ca profiles are remarkably similar among PBT otoliths in the general trend over the transect (Fig 3). Element:Ca data are grouped by region, but it is possible that PBT migrated across different regions throughout life. Li:Ca, Mg:Ca and Mn:Ca are elevated in core regions, then decrease to low values following the first (estimated) annuli position. Some specimens with peak Li:Ca and Mg:Ca near the otolith edge were removed as outliers (as discussed below). Highest Mn:Ca values occur in cores, with EPO slightly higher than WPO and SPO, but all specimens show a secondary peak between 800 to 1000 μm that rapidly decreases to near detection limits after ~age two (past 1500 μm). Note that all fish are assumed in the WPO spawning ground during the period that corresponds to the core. In contrast to Mn:Ca patterns, Zn:Ca, Sr:Ca and Ba:Ca are low in cores then show increasing trends after the second annuli that are differentiable by collection region (Fig 3). Zn:Ca gradually increase from core regions and plateau up to ages 1 to 3+, but increase again after 3000 μm. Sr:Ca show a small increase in the core that levels out through age-0 then further increases to 1800 μm at approximately age-3, before further increasing by region; continual increasing values characterize EPO and SPO, whereas lower stable values differentiate WPO. Ba:Ca diverges among capture regions after approximate age-3, with sharp, moderate, and much more gradual increases distinguishing the EPO, SPO, and WPO, respectively. In contrast to the other element:Ca ratios, otolith Ba:Ca patterns are highly oscillatory with high variability among individuals (S1 Fig), possibly reflecting differential movement among ocean regions prior to capture. Compared to EPO and SPO, oscillatory otolith Ba:Ca profiles have much lower amplitudes for WPO.

## Mean otolith edge patterns

The Grubbs test revealed 3 outliers (Li:Ca = 7.53 for EP05; Mg:Ca = 0.563 for NZ02, Mn:Ca = 1.66 for EP05) that were removed before the 1-way ANOVA test. Li:Ca, Mg:Ca, and Mn:Ca are not significantly different among regions, while Zn:Ca, Sr:Ca, and Ba:Ca are significantly different (Table 2; Fig 4). Holm-Sidak's multiple comparison tests indicate that otolith edge Zn:Ca was significantly higher in the SPO compared to the EPO (p = 0.02) and WPO (p = 0.04), but is not different between EPO and WPO (p = 0.52). Otolith edge Sr:Ca is significantly lower in the WPO versus EPO (p = 0.01) and SPO (p<0.001), but not different between the EPO and SPO (p = 0.07). The EPO has the highest otolith edge Ba:Ca, with significantly lower values in the WPO (p = 0.002), but EPO and SPO are not significantly different (p = 0.19) (Fig 4).

QDFA based on only Zn:Ca, Sr:Ca and Ba:Ca resulted in overall jackknifed classification success of 72% (Table 3). Classification success varies by each region, with 70% accuracy in the EPO with 1 misclassification occurring in the SPO and 2 in the WPO. The lowest classification success of 40% occurred in the SPO, with 3 misclassifications attributed to the EPO. WPO has the highest classification success of 90%, with only 1 misclassification in the EPO (Table 3). The regional classification differences were clear in the plot of the first and second discriminant scores (Fig 5). Two SPO fish clearly grouped together and were separated from the EPO and WPO, while 3 SPO fish more closely grouped to the EPO (Fig 5).

## Discussion

Reconstructing the life histories of migratory fishes requires tools that record endogenous and exogenous events experienced throughout ontogeny. This is the first study to investigate

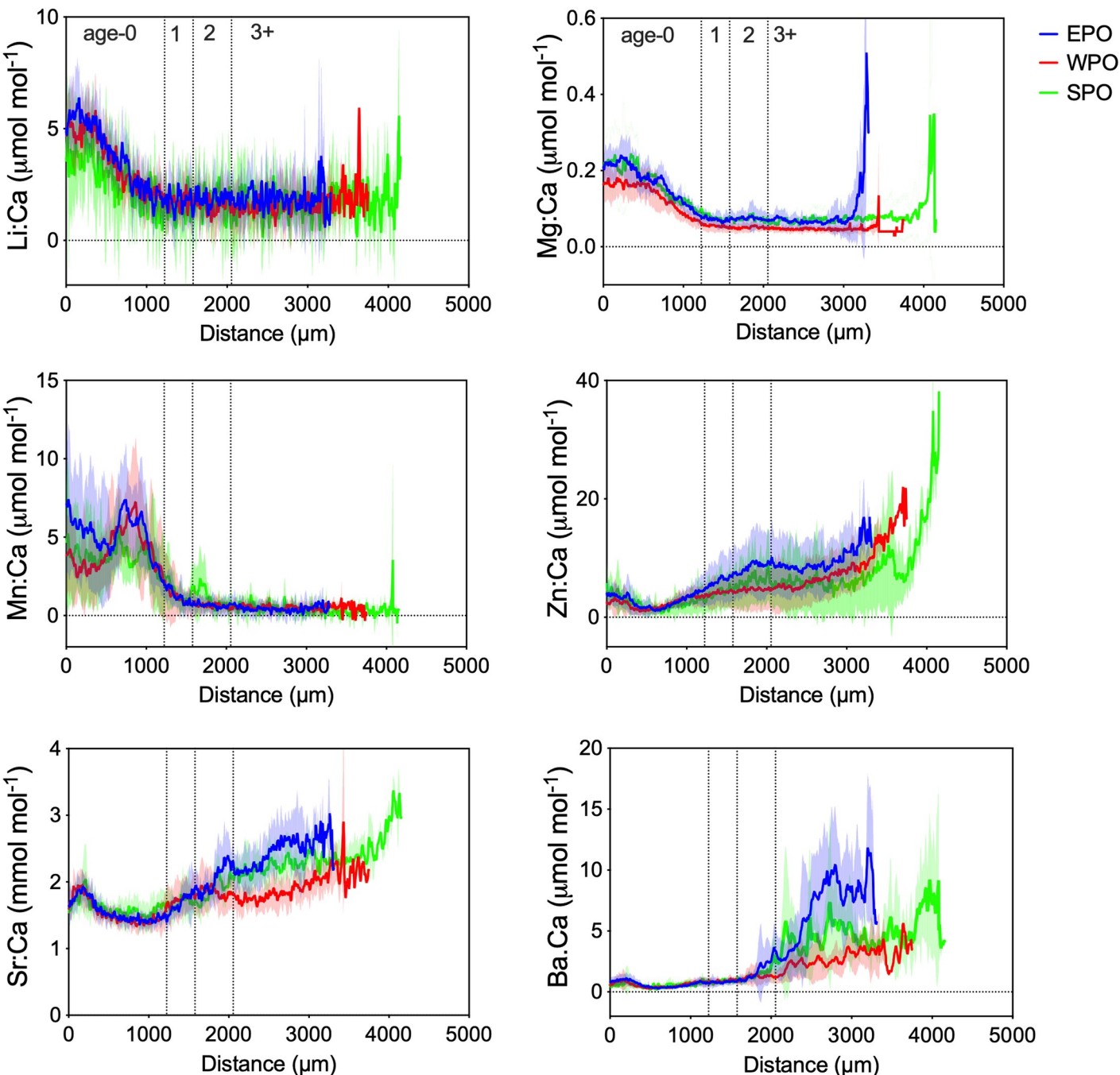

**Fig 3. Mean element:Ca profiles for otoliths of Pacific bluefin tuna.** (see Table 1 for fish collection details) collected from the eastern Pacific (EPO = blue, $N = 10$), western Pacific (WPO = red, $N = 10$), and south Pacific (SPO = green, $N = 5$) Ocean; solid lines represent mean values and shading depicts ± standard deviation across laser distance from core (0 μm) to edge; dashed vertical lines denote approximate annuli distances derived from [35].

otoliths of PBT collected from the SPO, a poorly understood migratory path where samples have rarely been collected or investigated. The element:Ca profiles of PBT otoliths collected from three regions (WPO, EPO, SPO) are similar during early life from age-0 to age-1 for all element:Ca ratios examined. This consistency suggests either 1) similar water mass occupancy during early life, 2) a strong physiological control on element incorporation in early life

**Table 2. One-way ANOVA results based on mean otolith edge (outer 100 µm) element:Ca values for Pacific bluefin tuna for exploring regional differences in otolith chemistry among the eastern Pacific (EPO), western Pacific (WPO) and south Pacific (SPO).** Only Mg:Ca was log transformed to meet normality assumption.

| Element | Factor | SS | DF | MS | F | P value |
|---|---|---|---|---|---|---|
| Li:Ca | Region | 1.38 | 2 | 0.6902 | 1.32 | 0.2883 |
| | Residual | 10.98 | 21 | 0.5228 | | |
| | Total | 12.36 | 23 | | | |
| log(Mg:Ca) | Region | 0.3879 | 2 | 0.194 | 3.335 | 0.0552 |
| | Residual | 1.221 | 21 | 0.05816 | | |
| | Total | 1.609 | 23 | | | |
| Mn:Ca | Region | 0.2734 | 2 | 0.1367 | 1.119 | 0.3452 |
| | Residual | 2.564 | 21 | 0.1221 | | |
| | Total | 2.838 | 23 | | | |
| Zn:Ca | Region | 408.6 | 2 | 204.3 | 4.642 | **0.0208** |
| | Residual | 968.2 | 22 | 44.01 | | |
| | Total | 1377 | 24 | | | |
| Sr:Ca | Region | 1.333 | 2 | 0.6664 | 10.73 | **0.0006** |
| | Residual | 1.366 | 22 | 0.0621 | | |
| | Total | 2.699 | 24 | | | |
| Ba:Ca | Region | 176 | 2 | 87.99 | 8.085 | **0.0023** |
| | Residual | 239.4 | 22 | 10.88 | | |
| | Total | 415.4 | 24 | | | |

independent of ambient water concentration [23, 36], or 3) a combination of both. This result is expected given that all fish originate from the WPO, however it is not possible to differentiate between the influence of regional water chemistry and physiological control. Wells et al. [29] found statistically significant differences in otolith core Mn:Ca, Mg:Ca, Sr:Ca and Zn:Ca between YOY fish from the East China Sea and Sea of Japan, suggesting at least these elements are influenced by regional conditions that vary interannually.

As juveniles, after the estimated age of 3, trends of three element:Ca profiles (Zn:Ca, Sr:Ca, and Ba:Ca) diverge by region of collection suggesting that environmental factors that vary in time and space (e.g., temperature and salinity) influence incorporation of those elements [24]. Thus, otolith element:Ca patterns characterized in this study align well with our current understanding of PBT life history, that includes a common spawning ground in the WPO with juveniles either remaining resident in the WPO or migrating to the EPO [4, 16, 17, 37] or larger fish moving into the SPO [14, 15].

Otolith geochemical records provide useful life history information for pelagic tuna species including natal origin determination [29, 38, 39], stock structure identification [38, 40, 41] and migrations with life history transects [30, 42–45]. Ontogenetic otolith chemistry patterns revealed by LA-ICP-MS core-to-edge transects in this study are similar to those derived from LA-ICP-MS discrete spot analyses [30] on PBT, as well as probe-based otolith studies of other migratory tuna species including south Pacific albacore *Thunnus alalunga* [42], skipjack tuna *Katsuwanus pelamus* [43] and southern bluefin tuna *Thunnus maccoyii* [44, 45]. Similarities in elemental profiles include higher Li and Mn in early life and general increases in Sr and Ba in older ages. Employing probe-based analysis to quantify continuous elemental patterns across sequential calcified and proteinaceous growth bands provides a chemical calendar revealing the life histories of highly migratory fish.

Identifying drivers of elemental variations during life requires understanding the relative influence of physiological and environmental processes on element incorporation and

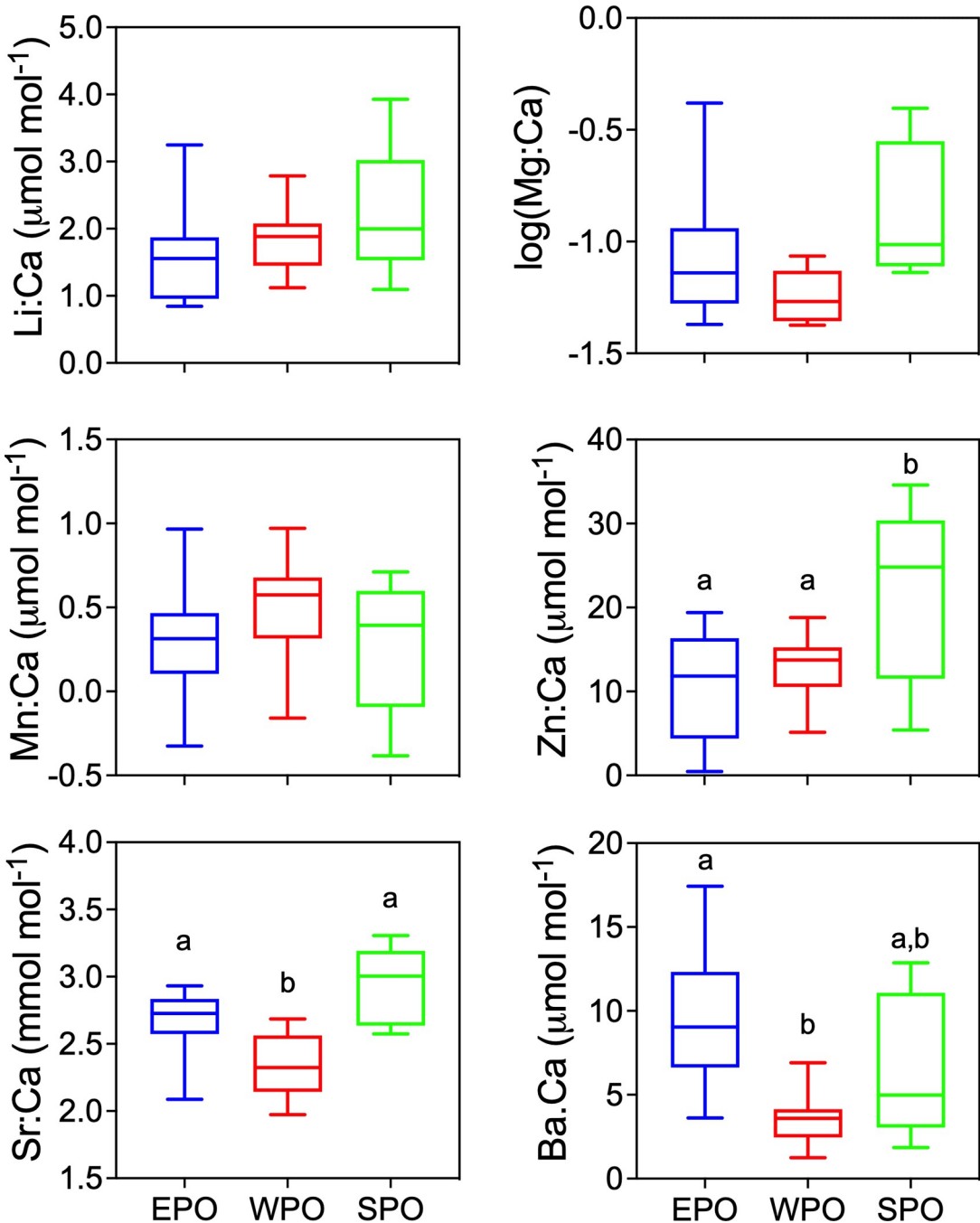

**Fig 4. Boxplots of otolith edge values (mean of final 100 μm) for each region.** EPO = blue, WPO = red, SPO = green. One-way ANOVA significant differences (see Table 1) among regions were further tested with a Holm-Sidak multiple comparison test with lower case letters indicating regions that are significantly different; regions that were not significantly different share the same lower case letter.

dissolved ion transfer from water and food, across gill and gut membranes, transport through blood-endolymph interfaces, and ultimately otolith biomineralization [24, 46]. Analytical limitations must also be recognized when interpreting laser scan profiles in biominerals. Optimizing laser setting (power (J/cm$^2$), repetition rate (Hz), scan rate (um s$^{-1}$) and ICP MS

**Table 3. Jackknifed classification success to the three regions (EPO, WPO, SPO) derived from quadratic discriminant function analysis based on otolith edge element:Ca values of Pacific bluefin tuna.**

| Known region of capture | Predicted region of capture | | | % correct classification |
|---|---|---|---|---|
| | EPO | SPO | WPO | |
| EPO | 7 | 1 | 2 | 70 |
| SPO | 3 | 2 | 0 | 40 |
| WPO | 1 | 0 | 9 | 90 |
| Total | 11 | 3 | 11 | 72 |

parameters (i.e. carrier gas (He, Ar) flow rate, cone distance, etc.) and using matrix-matched replicated standards is important for comparing studies conducted in different labs. Although laser spot diameters are standardized, crystal growth and accretion rates decrease with age [47], thus the laser integrates different time windows based on the age and growth rate of the fish. The geographic otolith edge comparison here integrated different time frames in the PBT since they differed in age, but early life otolith core comparisons should reflect similar growth and biomineral accretion rates in larval/juvenile PBT, regardless of collection location.

In PBT otolith cores, Li:Ca, Mg:Ca and Mn:Ca were enriched, then gradually decreased to low stable values after age 1. Physiological regulation of these elements have been shown in teleost fish [24, 48]. Thus, the enrichment of otolith Li:Ca, Mg:Ca and Mn:Ca, at early life may reflect periods of rapid juvenile growth rate, higher protein accumulation versus aragonite crystallization, and different metabolic rates before PBT exhibit endothermy. PBT undergo a

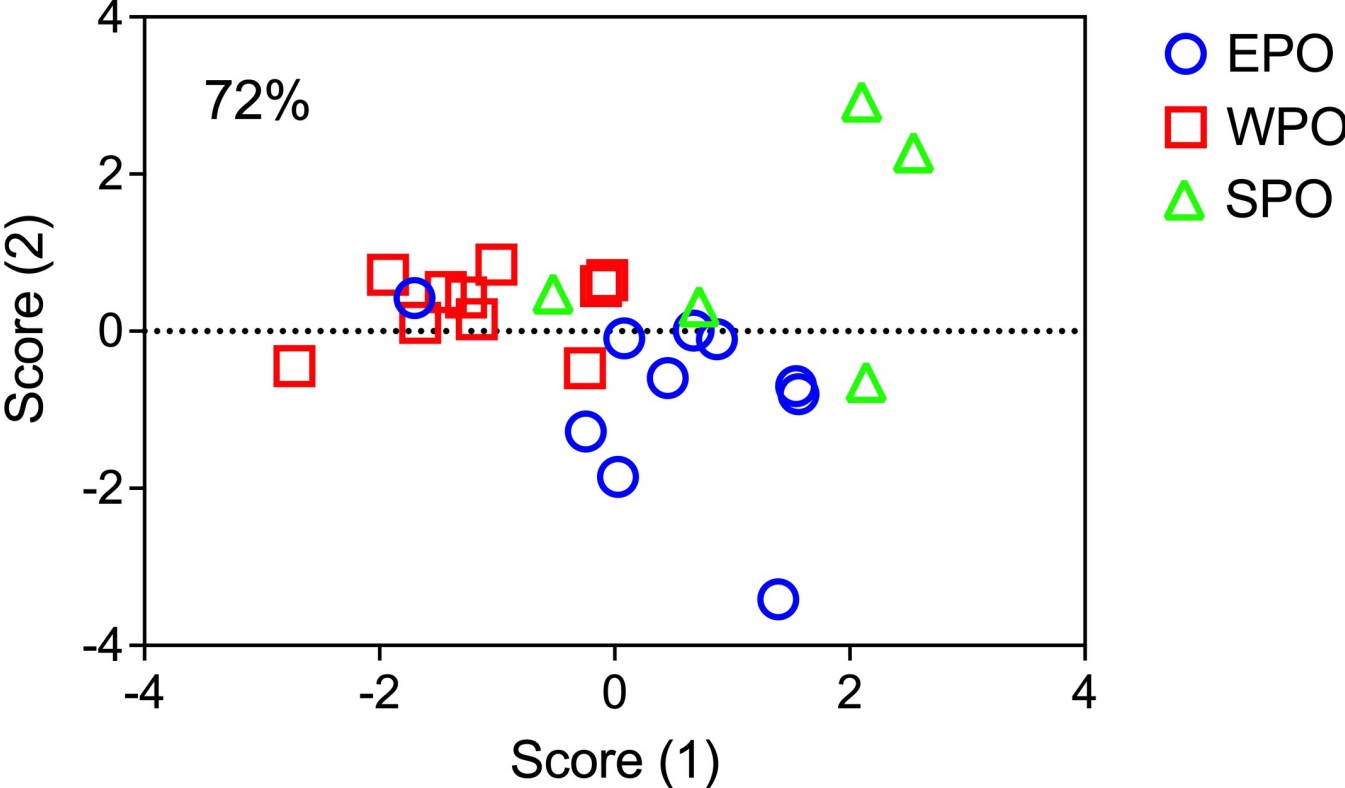

**Fig 5. Quadratic discriminant function analysis canonical score plot.** EPO = blue circle, WPO = red square, SPO = green triangle. The overall classification accuracy using Pacific bluefin tuna otolith edge Zn:Ca, Sr:Ca and Ba:Ca was 72%.

metamorphosis from larvae to juveniles at 20–35 days post hatch, with rapid increases in protein synthesis and somatic growth that correspond with drastic increases of protein-DNA and RNA-DNA ratios [49]. This period of rapid somatic growth would result in increased otolith accretion rates [50]. The peak of Mn:Ca within the core, but following the primordium within age-0, may correspond with the larvae-to-juvenile transition to rapid growth, where otolith morphology and growth axes shift. The fact that element concentrations change at this inflection point, where the aragonite growth axes also change direction, is consistent with ontogenetic changes in mineral growth or crystal formation as a control influencing elemental uptake. After the first 1000 μm (~age 1), core-enriched elements (Li:Ca, Mg:Ca and Mn:Ca) decrease and remain low throughout ontogeny.

Results from other studies suggest that Mn:Ca and Mg:Ca are also influenced by environmental conditions. These ratios were most useful for multivariate assignment of YOY PBT collected from two separate spawning regions using both solution- [27, 51] and LA-ICP-MS-based analysis of core regions [29]. In addition, Mn:Ca and Mg:Ca within the first 500 μm of core otolith differed inter-annually, which may reflect interannual environmental variability experienced by PBT spawned in different years (e.g., ambient water temperature and chemistry). PBT metabolic rates, foraging behavior, and prey availability within the two known spawning grounds, may also affect differential incorporation of Mn [52] and Mg [36] during early life. All of these factors can vary from year to year, so annual juvenile otolith chemistry baselines from each spawning region are essential for inferring natal origins of age-classed matched unknowns [29].

Our finding that Zn:Ca, Sr:Ca, and Ba:Ca have similar values in otolith cores but differ significantly in the last 100 μm of otolith edges among the three capture areas supports that these element:Ca ratios may proxy ocean basin-specific differences. The outer 100 μm of otolith growth likely represent $\geq$ 33 days in larger individuals with slower growth rates based on direct validation studies of southern bluefin tuna (*Thunnus maccoyii*) with estimated mean otolith accretion rates of 3 μm day$^{-1}$ [53]. For such time intervals it is reasonable that the specimens largely or completely resided within the capture regions, and thus associated element:Ca values were also obtained within capture regions.

Other studies have shown that otolith chemistry can distinguish among widely separated collection regions in the Pacific [40, 43]. For example, Arai et al. [43] found (EPMA-based) Sr profiles effective for distinguishing skipjack tuna migratory behavior between temperate and tropical waters, including a fish tagged in the WPO that migrated to the SPO and was recaptured [43]. For skipjack, Sr concentrations were lower in the cooler (19-22˚C) temperate waters (Japan coast) and higher in warmer (29-30˚C) tropical waters near the equator. Our finding that lowest PBT otolith edge Sr concentrations occur in specimens collected from the WPO, is consistent with the skipjack tuna patterns of Arai et al. [43], but we cannot confirm the specific temperature experiences of WPO fish were lower compared to EPO or SPO fish.

Based on differences in (EPMA-based) Na, Ca, Sr, S, K, and Cl profiles for PBT otolith, Proctor et al. [44] suggested that ontogenetic variability was greater than any environmental variability, and thus otolith chemistry would not be useful for delineating geographic stock structure due to the homogeneity of ocean chemistry for these conservative elements. However, injection of sea-cage PBT with SrCl$_2$ in the same study resulted in a strong spike in otolith concentrations, suggesting transfer of dissolved Sr$^{2+}$ from blood to otolith, but it is unknown how that experiment reflects natural conditions [20, 23, 54]. Sr$^{2+}$ substitution for Ca$^{2+}$ during calcium carbonate biomineralization is well documented and widely attributed to similar ionic radii [53, 55, 56]. That Sr$^{2+}$ is only found in the non-protein salt fraction of otoliths and not under physiological regulation [22], supports otolith Sr:Ca as an environmental proxy of water temperature and salinity [57], but may be complicated by biochemical, biological and

physiological interactions, including sexual maturation [20, 23, 54] and the development of increased endothermy with body size [58].

We find highest Ba concentrations in otoliths collected in the EPO, a well characterized upwelling region [59, 60]. Cold, nutrient rich upwelled water typically exhibits higher dissolved Ba concentrations than surface waters [61, 62]. Barium has been shown to proxy upwelling conditions in diverse calcified biominerals, including coral skeleton [63, 64], shark vertebrae [65] and fish otoliths [30, 40, 66]. Tagging studies demonstrate that PBT experience cold water temperatures when they occasionally dive below the thermocline, likely to forage on deep-water prey [67, 68]. EPO PBT mainly reside at the surface the water column [69], and move latitudinally over areas of coastal upwelling with high primary productivity [5]. The oscillations of otolith Ba:Ca detected after age 3 in our study, could follow movement into deeper, cooler water, when PBT forage on deep water prey and increased dissolved Ba is taken up through the gills or intestine. The width of oscillating Ba:Ca peaks are approximately seasonal (~300 μm) in transects of EPO and SPO otoliths. Ba:Ca amplitudes fluctuate between 10–20 μmol mol$^{-1}$ in these specimens, compared to amplitudes < 10 μmol mol$^{-1}$ in WPO specimens (see S1 Fig). These regional Ba:Ca patterns could reflect larger-scale seasonal movements to upwelling areas, as consistent with archival tagging data [5].

Highest otolith edge Zn concentrations correspond to SPO PBT, the region where the largest and oldest (3 PBT estimated 20+ years) fish were collected. Because Zn in seawater is commonly bound to organic complexes, dissolved $Zn^{2+}$ is not readily available for uptake compared to $Sr^{2+}$ and $Ba^{2+}$ [54, 70]. Previous studies indicate that $Zn^{2+}$ is under strong physiological control in fish, serving as co-factors in many enzymes and proteins [22, 54, 71]. Elevated Zn:Ca in otolith core regions, followed by decreasing levels with age, is a common ontogenetic pattern reported in other species [24]. Our data demonstrate an opposite pattern with Zn:Ca increasing with ontogeny and highest otolith edge Zn:Ca occurring in the oldest PBT. Increasing otolith Zn:Ca with age in PBT could indicate a physiological control, such as sexual maturity [54, 72] or reduced otolith accretion rate with age.

We found an overall average discrimination classification success of 72% among capture regions using Sr:Ca, Ba:Ca and Zn:Ca in the outer 100 μm of otolith edges. The limitation of using a constant otolith edge distance (100 μm) is that this distance will represent different time frames, equating to potentially years in the oldest PBT (7–20+ years in this study) as otolith increment growth slows down significantly with age in bluefin [73]. This outer-edge discrimination approach also assumes that the fish have been in capture regions long enough for local signatures to have been incorporated. All WPO PBT were collected in same month and year (May 2017), had similar estimated ages (range 9–13 y, mean ± standard deviation = 11 ±1.5 y) and exhibited the highest classification accuracy of 90% with only one misclassification. Thus, temporal variability of both collection location (all samples in one month) and otolith accretion rate was minimized for these specimens. The next highest classification accuracy was for EPO PBT at 70%, with two fish misclassified from WPO and one fish mistaken from SPO. The youngest PBT were also collected from the EPO, with a mean estimated age 7.2±0.7 years and collection dates within a year (July 2016 to Oct 2017). The lowest classification accuracy (40%) was obtained for SPO adults, likely due to wide ranging collection dates from 2007 to 2018, differences in fish size and age (6 to 20 y), and more disparate collection regions, including New Zealand and the Cook Islands separated by over 3,000 km in tropical waters, compared to EPO and WPO in temperate waters. The influence of interannual variability in oceanic conditions (i.e. temperature shifts for La Nina versus El Nino) on otolith biomineralization is a recognized factor degrading classification success using otolith geochemical signatures [74].

## Conclusion

This study examined if elemental time series of PBT otoliths reflect ocean basin-scale migratory movements or regional residence. Elemental transects of PBT otoliths provide ontogenetic records of physiological and environmental histories, although it is often difficult to discern between the two. Similarities of Li:Ca, Mg:Ca, and Mn:Ca profiles for juvenile PBT 1–2 years old (to 1500 µm distance from core) among all three geographically distinct capture regions are potentially related to similar thermal physiology, rapid growth and otolith accretion rates and a common region of origin within the WPO. After 2–3 years, Sr:Ca, Ba:Ca, and Zn:Ca begin to diverge by region of collection, likely reflecting spatial and temporal oceanographic variability experienced when PBT undertake broadscale migrations, or physiological influences associated with changes in foraging and/or breeding behavior. Without controlled laboratory experiments, which are very difficult for large bodied and fast-moving tunas, the relative influences water chemistry, ambient temperature, diet and metabolic physiology on otolith elemental uptake will be premised on descriptive studies. Additional research on captive reared PBT or otoliths of tagged and recaptured individuals will expand knowledge on elemental uptake in otolith biominerals in PBT. Future otolith geochemical studies involving a greater number of older (larger) specimens should further advance understanding PBT life history and migratory behaviors. Refining the otolith chemistry approach to characterize behavior including migratory and resident contingents, can help mangers better understand stock dynamics and improve stock assessment models for highly migratory species.

## Supporting information

**S1 Data.**
(XLSX)

**S1 Fig. Elemental profiles of individual PBT otoliths collected from the eastern Pacific (EPO), western Pacific (WPO) and south Pacific (SPO); different colored lines represent individual fish from each region.** See Table 1 for details on collection. Asterisk (*) indicates visible crystal bond on surface of NZ02 that was removed from Fig 1.
(PDF)

## Acknowledgments

We thank members of the National Institute of Far Seas Fisheries, National Institute of Water & Atmospheric Research (NIWA), Ministry for Primary Industries (New Zealand), and the SPC Pacific marine specimen tissue bank for assistance with Pacific bluefin tuna otoliths sample collections. In the EPO, PBT heads were generously donated by recreational sport fishermen, captains/crews, and fish processors.

## Author Contributions

**Conceptualization:** John A. Mohan, Heidi Dewar, Owyn E. Snodgrass, Yosuke Tanaka, Seiji Ohshimo, Jay R. Rooker, R. J. David Wells.

**Data curation:** John A. Mohan, Owyn E. Snodgrass, Nathan R. Miller, R. J. David Wells.

**Formal analysis:** John A. Mohan, Nathan R. Miller, R. J. David Wells.

**Funding acquisition:** John A. Mohan, Heidi Dewar, Jay R. Rooker, R. J. David Wells.

**Investigation:** John A. Mohan, Yosuke Tanaka, Seiji Ohshimo, Malcom Francis, R. J. David Wells.

**Methodology:** John A. Mohan, Owyn E. Snodgrass, Nathan R. Miller.

**Project administration:** John A. Mohan, Seiji Ohshimo, Jay R. Rooker, R. J. David Wells.

**Resources:** Malcom Francis, R. J. David Wells.

**Visualization:** John A. Mohan.

**Writing – original draft:** John A. Mohan, Heidi Dewar, Owyn E. Snodgrass, Nathan R. Miller, Yosuke Tanaka, Seiji Ohshimo, Jay R. Rooker, Malcom Francis, R. J. David Wells.

**Writing – review & editing:** John A. Mohan, Heidi Dewar, Owyn E. Snodgrass, Nathan R. Miller, Yosuke Tanaka, Seiji Ohshimo, Jay R. Rooker, Malcom Francis, R. J. David Wells.

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
