## [Decision Letter · Decision Letter 0]

3 Aug 2022

PONE-D-22-17432Otolith geochemistry reflects life histories of Pacific bluefin tunaPLOS ONE

Dear Dr. Mohan,

Thank you for submitting your manuscript to PLOS ONE. After careful consideration, we feel that it has merit but does not fully meet PLOS ONE’s publication criteria as it currently stands. The reviewers recommend minor revisions. Therefore, we invite you to submit a revised version of the manuscript that addresses the points raised during the review process. Please submit your revised manuscript by Sep 17 2022 11:59PM. If you will need more time than this to complete your revisions, please reply to this message or contact the journal office at plosone@plos.org. Please include the following items when submitting your revised manuscript:A rebuttal letter that responds to each point raised by the academic editor and reviewer(s). You should upload this letter as a separate file labeled 'Response to Reviewers'.A marked-up copy of your manuscript that highlights changes made to the original version. You should upload this as a separate file labeled 'Revised Manuscript with Track Changes'.An unmarked version of your revised paper without tracked changes. You should upload this as a separate file labeled 'Manuscript'.

We look forward to receiving your revised manuscript.

Kind regards,

Antonio Medina Guerrero, Ph.D.

Academic Editor

PLOS ONE

Journal Requirements:

2. In your manuscript, please provide additional information regarding the specimens used in your study. Ensure that you have reported specimen numbers and complete repository information, including museum name and geographic location.

"This work was funded by a grant from the NOAA Saltonstall-Kennedy research program (grant # NA17NMF4270224)."

Reviewers' comments:

Reviewer's Responses to Questions

**Comments to the Author**

1. Is the manuscript technically sound, and do the data support the conclusions?

Reviewer #1: Yes

Reviewer #2: Yes

2. Has the statistical analysis been performed appropriately and rigorously? 

Reviewer #1: Yes

Reviewer #2: Yes

3. Have the authors made all data underlying the findings in their manuscript fully available?

Reviewer #1: Yes

Reviewer #2: Yes

4. Is the manuscript presented in an intelligible fashion and written in standard English?

Reviewer #1: Yes

Reviewer #2: Yes

5. Review Comments to the Author

Reviewer #1: This study presents an original research article on otolith trace element profile analysis of Pacific bluefin tuna (Thunnus orientalis), as an approach to provide records of their life history. Authors apply for the first time this technique in Pacific bluefin tuna collected across the species adult distributional range; covering distant areas from the Pacific Ocean (i.e., west, east, and south) and discuss on how different elements provide ontogenetic records of physiological and environmental histories.

Overall, the article is presented intelligibly and is easy to follow. Both experiments and statistics are performed to a high technical standard and are described in sufficient detail, so that other scientist can follow this study. Authors also state that data is available under request without restrictions. Results are rigorously reported and support the conclusions of the article. The motivation behind the work is clearly described, with straight and sound discussion of the observed results and in the limitations of the performed study.

I do believe that this research meets all criteria for publication in PLOS ONE journal. However, there are some minor issues that could be improved in my point of view before its publication as a high-quality manuscript:

- My biggest concern is with the readability of the figures and tables. I don't know if it is a problem with the system, or with the quality of the figures themselves, but they should be thoroughly revised as they are difficult to read as I see them. Please check them for quality standards and size format.

- I think the discussion lacks a little bit more, the microchemistry part is very well detailed and contextualized, but it seems to me a good opportunity to discuss a little bit more about migratory behaviours, contingents and discuss some implications of these findings for the Pacific bluefin tuna management and fisheries as well as and future research priorities. Besides, the introduction of the discussion needs to be refined a bit by looking at the connection between the ideas to be presented later.

If authors can address these minor issues, I do believe that this manuscript will be a great addition to the PLOS ONE articles collection and to the scientific community working with otolith or tuna management related studies. Bellow I have provided specific comments and/or questions to the authors that should be quite straightforward to address. I hope authors and editor can find these suggestions useful, and my congratulations for this nice piece of work, this is good science that needs to be shared!

Introduction

Line 85: I suggest using “core-to-edge” as it is more common in the literature and then in MS you are referring to that part as “edge”.

Lines 83-96: I would add also a small sentence saying that there is also substantial uncertainty on how intrinsic and extrinsic processes affect elemental incorporation into the otolith, and also that responses can be species specific (e.g. Hüssy et al., 2020; Sturrock et al., 2015).

Line 93: You can cite Farwell et al. 2001 at the end of this sentence if you want, which says “The open ocean, pelagic environment of most scombrid species has made it challenging to conduct field research. Their high metabolic rates and specialized swimming result in significant space requirements if held in captivity. The major challenge is that tunas must swim continuously, which makes collecting and captive care difficult”.

Farwell, C. J. (2001). 10. Tunas in captivity. Fish physiology, 19, 391-412.

Line 99: And differentiate among contingents or fish with different life histories?

Line 108-109: I suggest replacing “LA ICP MS” by “LA-ICP-MS”

Line 109: From hatch time to adult?

Line 110: I feel that “comparing complete chemical profiles” would be more accurate than “comparing complete life histories”

Line 114: I suggest deleting “biominerals” at the end of the sentence.

Material and Methods

Lines 118-120: I think it will be nice to indicate here how many otoliths did you get from each region, and not later in Results section.

Line 120: I feel that a map would be a great added value here, where you can show the location of the 3 regions you are considering (instead of figure S1), and the location of the spawning areas. I think this will be a plus for the manuscript and will help the non PBT familiarized readers to locate the regions you mention throughout the manuscript (I needed to check in google maps to get the whole picture in my mind).

Table 1: I think that table 1 needs from some restructuration for the paper, for example, I am not sure if both PBT_ID and Otolith ID are relevant for the reader (perhaps choose only one). Same I am not sure if the fishing methods adds something here? Like this you could use the space you have left over to incorporate a “Region” column where you indicate if samples belonged to WPO, EPO or SPO a with the Location only, readers can be confused (as it was my case because I am not familiarized with the Pacific). Please add to the legend that size is fork length (FL). Also check that are some measures of FL and GG that are in bold.

Lines 129-130: “It was also impossible to know with certainty the migratory histories of sampled fish…” sure, but this is what you want to inspect no? I would suggest deleting this sentence and said only that fish were grouped by sampling region for comparisons.

Figure 1b: Perhaps it is worthy to write core, inflexion point and edge in the figure to help the readers locate the otolith position afterwards? Not sure…

Line 151. Crystalbond “adhesive” or “glue” instead of “cement”?

Line 165: Perhaps “Otolith sections and standards…”

Line 173: Perhaps you can cite Sturgeon et al., 2005 at the end of this sentence.

Sturgeon, R. E., Willie, S. N., Yang, L., Greenberg, R., Spatz, R. O., Chen, Z., ... & Thorrold, S. (2005). Certification of a fish otolith reference material in support of quality assurance for trace element analysis. Journal of Analytical Atomic Spectrometry, 20(10), 1067-1071.

Line 190: Are these mean annual annuli? If so please state it in the text

Line 190: But this image only shows annual rings up to 3 years, how did you do for your older individuals?

Line 197: I know you mention later in the discussion, which is really nice, but I think it would also be worthy to mention here that time frame represented in each individual is different.

Line 200-201: Was your data normally distributed and present homogeneity of variances so that it justifies the use of one way ANOVA? If so please state, if not, consider using another non-parametric test. I see that you mention latter that a Brown- Forsythe test was performed and confirmed that variances of the populations from which the samples are drawn are equal, but what about normal distribution?

Results

Lines 212-216: I am not sure whether this should go here or in M&M, I feel that it does not fit in results, and it fits better in M&M sampling description.

Line 219: “...but it is possible that PBT…” ?

Line 236: Perhaps add “ratios” after “element:Ca”

Figure 2: “solid line represent mean values and shading depicts ± standard deviation across laser distance” I cannot see this in the figure I get. Please check just in case. “dashed vertical lines denote approximate annuli distances for the first 3 years of life…”, I think it will also be beneficial to add where the inflection point will be represented in the transect, as then you mention this in you discussion.

Figure 3: I personally found confusing the fact that horizontal barckets join regions that are not statistically significant, when usually it is the other way around, I would just mark between which there are significant differences.

Table 2: This table need a little bit of restructuring and editing. Please also indicate in the legend that significant P values are highlighted in bold in the table. Also add that inspected regional differences are between WPO, EPO and SPO.

Lines 282-283: I suggest expanding here and also to mention that although classification differences are more or less clear for WPO and EPO there is also some overlap. Also, that SPO is disperse, and not clear grouping can be observed. So, there are 2 SPO with similar values among them, 1 more “EPO like” and 2 more “WPO like”, as you can see in the classification of table 3.

Table 3: I recommend replacing “origin” by “capture” here. Perhaps you can highlight in bold the correct classifications of the table.

Discussion

Lines 296-297: I feel that this sentence is a bit weak as it is, and you should contextualise it more.

Lines 295-307: I think that some cohesion in the first paragraph of the discussion is missing.

Line 303: All of your fish are originated from the WPO no? not only the juveniles

Line 318: I suggest replacing “core-to-rim” by “core-to-edge”

Lines 318-322: Are all of these in the Pacific? Can you specify more on what similarities did you found? (e.g., enrichment/depletion for early life signatures etc.

Lines 322-324: And also continuous transects no?

Lines 325-333: I think you could also discuss here on the importance of the placement of the transect, as some elements do not precipitate equally in the otolith growth axis (you can check Artetxe-Arrate et al. 2021)

Artetxe-Arrate, I., Fraile, I., Clear, N., Darnaude, A. M., Dettman, D. L., Pécheyran, C., ... & Murua, H. (2021). Discrimination of yellowfin tuna Thunnus albacares between nursery areas in the Indian Ocean using otolith chemistry. Marine Ecology Progress Series, 673, 165-181.

Line 365: What individual variability do you see in you fish for early life period of these elements as you are accounting for fish that were born at different years? Do you see some fish with signatures more similar to the East China Sea and other to Sea of Japan reported in Wells et al. 2020? Maybe you can discuss a bit more here

Line 382: Are lower temperatures expected for the WPO than for EPO and SPO? If so please state it

Line 402-404: But are these occasional dives long enough to be recorded in the chemical signature? I don’t think so…

Lines 419-424: We also found that otolith borders were enriched in Zn in comparison with the rest of the otolith for few different tuna species, perhaps because of Crystalbond inclusion, or because the otolith crystal composition of the border. I don’t know if it is worthy to also consider this in small sentence here?

I do miss a little discussion on contingents and migratory behaviour, and also to recap the importance of this knowledge for management (following your intro)

Conclusions

Perhaps recap your objectives and how you answer them with this study

Lines 448-449: Perhaps you can add “although it is often difficult to discern between the two”

Lines 461: And in other fish in general, as these type studies are mostly lacking for any species.

Best luck.

Reviewer #2: Mohan et al. used otolith geochemical transects (core-to-edge) to reconstruct life histories of adult Pacific Bluefin tuna (PBT). They analyzed trace elements in otoliths of adult PBT obtained from the western, eastern, and southern Pacific Ocean to investigate patterns across ontogeny. Results show that some elements (Li:Ca, Mg:Ca, Mn:Ca) were higher during the first 1-2 years and then decreased independently of fish capture location, while other elements (Sr:Ca, Ba:Ca, Zn:Ca) showed a similar pattern in the first 1-2 years but then showed varying patterns likely related to different migration pathways to the capture location. Overall, the manuscript is well written and showcases the utility of otolith geochemical signatures to investigate entire life histories (birth to capture). I have only a few comments that I would like the authors to address before the manuscript can be accepted for publication in Plos One.

The authors only describe the otolith elemental transect data and do not perform any formal statistical testing. It could be worthwhile to consider using a time-series clustering approach analyzing the first 1-2 years and the remaining years to help support the statement that all PBT spawning and nursery areas are located in the EPO and that after ~2 years PBT migrate to different areas.

Using 100 um from the edge as the capture location signature can be very misleading as you discuss in the Discussion section. Why didn´t you consider using individual age-related distance to obtain a edge signature that corresponds roughly to the same time frame in each individual?

Table 1 legend: Change to “Pacific bluefin tuna (PBT) metadata including collection date, region and method;…”

Table1: Why some of the number are in bold?

Line 144: “Otoliths of PBT were sectioned…”

Line 192-193: Please clarify this sentence. As I understand it, you are referring to the distance from the core to age 1, age 2 etc, but can be interpreted as distance between annuli.

Line 357: YOY PBT

Figures 2-4: Need to be substantially improved for publication.

6. PLOS authors have the option to publish the peer review history of their article (what does this mean?). If published, this will include your full peer review and any attached files.

Reviewer #1: No

Reviewer #2: No

---

## [Author Response · Author response to Decision Letter 0]

22 Sep 2022

Mohan Response to reviewers comments

Review Comments to the Author

This study presents an original research article on otolith trace element profile analysis of Pacific bluefin tuna (Thunnus orientalis), as an approach to provide records of their life history. Authors apply for the first time this technique in Pacific bluefin tuna collected across the species adult distributional range; covering distant areas from the Pacific Ocean (i.e., west, east, and south) and discuss on how different elements provide ontogenetic records of physiological and environmental histories.

Overall, the article is presented intelligibly and is easy to follow. Both experiments and statistics are performed to a high technical standard and are described in sufficient detail, so that other scientist can follow this study. Authors also state that data is available under request without restrictions. Results are rigorously reported and support the conclusions of the article. The motivation behind the work is clearly described, with straight and sound discussion of the observed results and in the limitations of the performed study.

I do believe that this research meets all criteria for publication in PLOS ONE journal. However, there are some minor issues that could be improved in my point of view before its publication as a high-quality manuscript:

My biggest concern is with the readability of the figures and tables. I don't know if it is a problem with the system, or with the quality of the figures themselves, but they should be thoroughly revised as they are difficult to read as I see them. Please check them for quality standards and size format.

Response: The figure files were uploaded following the specific requirements of PLOS PONE for PRISM: 

Prism: Export your graph with the following settings: File format: TIFF; Resolution: 300; Color Mode: RGB; Size Make Width: 7.5 in; Enable Compression

I think the discussion lacks a little bit more, the microchemistry part is very well detailed and contextualized, but it seems to me a good opportunity to discuss a little bit more about migratory behaviours, contingents and discuss some implications of these findings for the Pacific bluefin tuna management and fisheries as well as and future research priorities. Besides, the introduction of the discussion needs to be refined a bit by looking at the connection between the ideas to be presented later.

Response: Thanks we have added lines 530 in conclusion: “Refining the otolith chemistry approach to characterize behavior including migratory and resident contingents, can help mangers better understand stock dynamics and improve stock assessment models for highly migratory species.”

If authors can address these minor issues, I do believe that this manuscript will be a great addition to the PLOS ONE articles collection and to the scientific community working with otolith or tuna management related studies. Bellow I have provided specific comments and/or questions to the authors that should be quite straightforward to address. I hope authors and editor can find these suggestions useful, and my congratulations for this nice piece of work, this is good science that needs to be shared!

Response: Thanks!

Introduction

Line 85: I suggest using “core-to-edge” as it is more common in the literature and then in MS you are referring to that part as “edge”.

Response: Done 

Lines 83-96: I would add also a small sentence saying that there is also substantial uncertainty on how intrinsic and extrinsic processes affect elemental incorporation into the otolith, and also that responses can be species specific (e.g. Hüssy et al., 2020; Sturrock et al., 2015).

Response: Added on line 92 ” There is high uncertainty on the relative influence of intrinsic and extrinsic processes that affect element incorporation into fish otolith and responses are species specific (23). “

Line 93: You can cite Farwell et al. 2001 at the end of this sentence if you want, which says “The open ocean, pelagic environment of most scombrid species has made it challenging to conduct field research. Their high metabolic rates and specialized swimming result in significant space requirements if held in captivity. The major challenge is that tunas must swim continuously, which makes collecting and captive care difficult”. 

Farwell, C. J. (2001). 10. Tunas in captivity. Fish physiology, 19, 391-412.

Response: Done 

Line 99: And differentiate among contingents or fish with different life histories?

Response: Added line 102 ” …identification of migratory contingents.”

Line 108-109: I suggest replacing “LA ICP MS” by “LA-ICP-MS”

Response: Done 

Line 109: From hatch time to adult?

Response: We prefer using juvenile to adult life. Hatch time would infer the laser could resolve larval stages, but the laser spot of 50 microns integrates on larger ‘weekly’ time scales and thus using juvenile stage is more appropriate.

Line 110: I feel that “comparing complete chemical profiles” would be more accurate than “comparing complete life histories”

Response: Changed to “By comparing complete elemental profiles of PBT sourced from different ocean basins, this study explores…”

Line 114: I suggest deleting “biominerals” at the end of the sentence.

Response: Done 

Material and Methods

Lines 118-120: I think it will be nice to indicate here how many otoliths did you get from each region, and not later in Results section.

Response: Added samples sizes to each region as suggested.

Line 120: I feel that a map would be a great added value here, where you can show the location of the 3 regions you are considering (instead of figure S1), and the location of the spawning areas. I think this will be a plus for the manuscript and will help the non PBT familiarized readers to locate the regions you mention throughout the manuscript (I needed to check in google maps to get the whole picture in my mind). 

Response: New Figure 1 added from supplement to manuscript.

Table 1: I think that table 1 needs from some restructuration for the paper, for example, I am not sure if both PBT_ID and Otolith ID are relevant for the reader (perhaps choose only one). Same I am not sure if the fishing methods adds something here? Like this you could use the space you have left over to incorporate a “Region” column where you indicate if samples belonged to WPO, EPO or SPO a with the Location only, readers can be confused (as it was my case because I am not familiarized with the Pacific). Please add to the legend that size is fork length (FL). Also check that are some measures of FL and GG that are in bold. 

Response: Thank you for the suggestions. Only one PBT ID is now presented. The fishing method was removed, and Ocean region of collection column added. The bold font was also removed, and legend indicated the FL=fork length

Lines 129-130: “It was also impossible to know with certainty the migratory histories of sampled fish…” sure, but this is what you want to inspect no? I would suggest deleting this sentence and said only that fish were grouped by sampling region for comparisons.

Response: Deleted and revised as suggested: lines 143: “Region of collection (EPO, WPO, SPO) was used to compare elemental patterns.”

Figure 1b: Perhaps it is worthy to write core, inflexion point and edge in the figure to help the readers locate the otolith position afterwards? Not sure…

Response: Thanks for suggestion. Added additional labels to new Figure 2b.

Line 151. Crystalbond “adhesive” or “glue” instead of “cement”?

Response: Changed “cement” to “adhesive” as suggested

Line 165: Perhaps “Otolith sections and standards…”

Response: Changed as suggested.

Line 173: Perhaps you can cite Sturgeon et al., 2005 at the end of this sentence.

Sturgeon, R. E., Willie, S. N., Yang, L., Greenberg, R., Spatz, R. O., Chen, Z., ... & Thorrold, S. (2005). Certification of a fish otolith reference material in support of quality assurance for trace element analysis. Journal of Analytical Atomic Spectrometry, 20(10), 1067-1071.

Response: Since we did not use a certified otolith standard, we did not include this citation. We used a calcium carbonate standard MACS-3 as the primary reference standard, not a certified otolith standard.

Line 190: Are these mean annual annuli? If so please state it in the text

Response: Yes, mean annuli distances are stated in the text line 215: “The mean annuli distances were 1,223 µm in year one, 1,577 µm in year two, 2,054 µm for year 3+.”

Line 190: But this image only shows annual rings up to 3 years, how did you do for your older individuals?

Response: We only calculated mean annuli distance for the first 3 years, when the annuli are widely spaced. Older ages and annuli distances were not estimated because data was not available. In Pacific bluefin tuna, the annuli become tightly spaced together at older ages, thus mean distances would not be accurate or comparable among individuals.

Line 197: I know you mention later in the discussion, which is really nice, but I think it would also be worthy to mention here that time frame represented in each individual is different.

Response: Added line 221: “For each otolith, the final 100 µm of the otolith edge was averaged to represent the recent otolith material accreted in the region of collection, which represents different time frames for individual fish.”

Line 200-201: Was your data normally distributed and present homogeneity of variances so that it justifies the use of one way ANOVA? If so please state, if not, consider using another non-parametric test. I see that you mention latter that a Brown- Forsythe test was performed and confirmed that variances of the populations from which the samples are drawn are equal, but what about normal distribution? 

Response: Thanks for suggesting testing the normality assumption. Only one element was not normal and thus was log transformed and then meet the normality assumption. Added lines 223: “Normality was assessed using a Kologorov-Smirnov test and only Mg was not normal distributed, and thus a log transform was used to meet normality assumption for Mg.”

Results

Lines 212-216: I am not sure whether this should go here or in M&M, I feel that it does not fit in results, and it fits better in M&M sampling description.

Response: Coauthors felt this was a result, so we have detailed the otolith collection sample sizes in both the M&M and results

Line 219: “...but it is possible that PBT…” ?

Response: Yes, thank you. “that” was added to text.

Line 236: Perhaps add “ratios” after “element:Ca”

Response: Done

Figure 2: “solid line represent mean values and shading depicts ± standard deviation across laser distance” I cannot see this in the figure I get. Please check just in case. “dashed vertical lines denote approximate annuli distances for the first 3 years of life…”, I think it will also be beneficial to add where the inflection point will be represented in the transect, as then you mention this in you discussion.

Response: New Figure 3 (previously Figure 2) been updated in full resolution to see details. It is likely the poor resolution of the original figure obscured shading of the standard deviation. We are unable to add the inflection point, since it differs between each individual fish otolith and this figure represents mean concentrations grouping fish by region. 

Figure 3: I personally found confusing the fact that horizontal barckets join regions that are not statistically significant, when usually it is the other way around, I would just mark between which there are significant differences. 

Response: Removed brackets that connected non-significant differences. Used lower case letter above groups that did display significant multiple comparison test, with non-significant groups sharing the same letter in new Figure 4 (previously Fig 3).

Table 2: This table need a little bit of restructuring and editing. Please also indicate in the legend that significant P values are highlighted in bold in the table. Also add that inspected regional differences are between WPO, EPO and SPO. 

Response: Table 2 has been updated to reflect the log(MgCa) results and the legend has been updated: “One-way ANOVA results based on mean otolith edge (outer 100 �m) element:Ca values for Pacific bluefin tuna for exploring regional differences in otolith chemistry among the Eastern Pacific (EPO), Western Pacific (WPO) and South Pacific (SPO). Only Mg:Ca was log transformed to meet normality assumption.”

Lines 282-283: I suggest expanding here and also to mention that although classification differences are more or less clear for WPO and EPO there is also some overlap. Also, that SPO is disperse, and not clear grouping can be observed. So, there are 2 SPO with similar values among them, 1 more “EPO like” and 2 more “WPO like”, as you can see in the classification of table 3.

Response: Added line 330: “Two SPO fish clearly grouped together and were separated from the EPO and WPO, while 3 SPO fish more closely grouped to the EPO (Fig 5).

Table 3: I recommend replacing “origin” by “capture” here. Perhaps you can highlight in bold the correct classifications of the table.

Response: Thank you. New table 3 updated.

Discussion

Lines 296-297: I feel that this sentence is a bit weak as it is, and you should contextualise it more.

Response: Added lines 351: “This is the first study to investigate otoliths of PBT collected from the SPO, a poorly understood migratory path where samples have rarely been collected or investigated.”

Lines 295-307: I think that some cohesion in the first paragraph of the discussion is missing.

Response: The first paragraph of discussion is providing a summary of overall results and linking to recent related work of Wells et al. 2020.

Line 303: All of your fish are originated from the WPO no? not only the juveniles

Response: Removed “juveniles”

Line 318: I suggest replacing “core-to-rim” by “core-to-edge”

Response: Done

Lines 318-322: Are all of these in the Pacific? Can you specify more on what similarities did you found? (e.g., enrichment/depletion for early life signatures etc.

Response: Added line 379: “Similarities in elemental profiles include higher Li and Mn in early life and general increases in Sr and Ba in older ages.”

Lines 322-324: And also continuous transects no?

Response: Yes, probe-based analysis refers to continuous transects.

Lines 325-333: I think you could also discuss here on the importance of the placement of the transect, as some elements do not precipitate equally in the otolith growth axis (you can check Artetxe-Arrate et al. 2021)

Artetxe-Arrate, I., Fraile, I., Clear, N., Darnaude, A. M., Dettman, D. L., Pécheyran, C., ... & Murua, H. (2021). Discrimination of yellowfin tuna Thunnus albacares between nursery areas in the Indian Ocean using otolith chemistry. Marine Ecology Progress Series, 673, 165-181.

Response: While we agree this is important, we placed the laser path on the same otolith growth axis for each individual. We would only be able to assess spatial variable with a 2-dimension elemental map, which is not available for these samples.

Line 365: What individual variability do you see in you fish for early life period of these elements as you are accounting for fish that were born at different years? Do you see some fish with signatures more similar to the East China Sea and other to Sea of Japan reported in Wells et al. 2020? Maybe you can discuss a bit more here

Response: Thank you for the comment. Unfortunately, we are unable to assess this as baseline YOY samples from the birth years of these fish are not available. Matched YOY baselines would be needed to address this, since interannual variability of juvenile baselines can affect interpretation of the natal chemical signatures.

Line 382: Are lower temperatures expected for the WPO than for EPO and SPO? If so please state it

Response: Thank you for the comment. We are not able to assess the temperature since we cannot define exact locations of the ocean that fish experienced, since they are highly migratory but also regionally endothermic. Added: lines 442 ”Our finding that lowest PBT otolith edge Sr concentrations occur in specimens collected from the WPO, is consistent with the skipjack tuna patterns of Arai et al. (2005), but we cannot confirm the specific temperature experiences of WPO fish were lower compared to EPO or SPO.”

Line 402-404: But are these occasional dives long enough to be recorded in the chemical signature? I don’t think so…

Response: Good point. However, if dives below the thermocline are consistent over several weeks occurring daily, then the integrated otolith signature might reflect that accumulative diving behaviour. 

Lines 419-424: We also found that otolith borders were enriched in Zn in comparison with the rest of the otolith for few different tuna species, perhaps because of Crystalbond inclusion, or because the otolith crystal composition of the border. I don’t know if it is worthy to also consider this in small sentence here?

Response: Thank you for this information. Since we did not see a similar Zn enrichment near the edge, we prefer to not discuss here.

I do miss a little discussion on contingents and migratory behaviour, and also to recap the importance of this knowledge for management (following your intro)

Response: Added lines 530: “Refining the otolith chemistry approach to characterize behavior including migratory and resident contingents, can help mangers better understand stock dynamics and improve stock assessment models for highly migratory species.”

Conclusion

Perhaps recap your objectives and how you answer them with this study

Response: Added sentence line 513: “This study examined if elemental time series of PBT reflect ocean basin-scale migratory movements or regional residence.”

Lines 448-449: Perhaps you can add “although it is often difficult to discern between the two”

Response: Added lines 515: “Elemental transects of PBT otoliths provide ontogenetic records of physiological and environmental histories, although it is often difficult to discern between the two” 

Lines 461: And in other fish in general, as these type studies are mostly lacking for any species.

Response: True, but this is especially true for large pelagic species that cannot be kept in captivity, such as the PBT studies here. 

Best luck.

Reviewer #2: Mohan et al. used otolith geochemical transects (core-to-edge) to reconstruct life histories of adult Pacific Bluefin tuna (PBT). They analyzed trace elements in otoliths of adult PBT obtained from the western, eastern, and southern Pacific Ocean to investigate patterns across ontogeny. Results show that some elements (Li:Ca, Mg:Ca, Mn:Ca) were higher during the first 1-2 years and then decreased independently of fish capture location, while other elements (Sr:Ca, Ba:Ca, Zn:Ca) showed a similar pattern in the first 1-2 years but then showed varying patterns likely related to different migration pathways to the capture location. Overall, the manuscript is well written and showcases the utility of otolith geochemical signatures to investigate entire life histories (birth to capture). I have only a few comments that I would like the authors to address before the manuscript can be accepted for publication in Plos One.

The authors only describe the otolith elemental transect data and do not perform any formal statistical testing. It could be worthwhile to consider using a time-series clustering approach analyzing the first 1-2 years and the remaining years to help support the statement that all PBT spawning and nursery areas are located in the EPO and that after ~2 years PBT migrate to different areas.

Response: Thanks for the suggestion. Unfortunately, an additional time-series clustering approach is beyond the scope of this study. We feel the current graphs and statistical approach of comparing otolith edge regions sufficiently support the conclusion of the study.

Using 100 um from the edge as the capture location signature can be very misleading as you discuss in the Discussion section. Why didn´t you consider using individual age-related distance to obtain a edge signature that corresponds roughly to the same time frame in each individual?

Response: Since the ages were only estimated based on size and not based on individual annuli counts of these tuna otoliths, our approach of standardized edge distance is the best we can do. We clearly discuss the limitations and caveats of our approach.

Table 1 legend: Change to “Pacific bluefin tuna (PBT) metadata including collection date, region and method;…”

Response: Done

Table1: Why some of the number are in bold?

Response: Corrected this error.

Line 144: “Otoliths of PBT were sectioned…”

Response: Thank you for catching this. Corrected.

Line 192-193: Please clarify this sentence. As I understand it, you are referring to the distance from the core to age 1, age 2 etc, but can be interpreted as distance between annuli.

Response: As stated, this is estimated annuli distance based on the laser path direction, not distance between annuli. Added: “The mean annuli distances were 1,223 µm in year one, 1,577 µm in year two, 2,054 µm for year 3+, as measured from the otolith core and along the laser path.”

Line 357: YOY PBT

Response: Thank you. Corrected.

Figures 2-4: Need to be substantially improved for publication.

Response: Yes, agreed. Higher resolution images will be uploaded with revised figures.

---

## [Decision Letter · Decision Letter 1]

27 Sep 2022

Otolith geochemistry reflects life histories of Pacific bluefin tuna

PONE-D-22-17432R1

Dear Dr. Mohan,

We’re pleased to inform you that your manuscript has been judged scientifically suitable for publication and will be formally accepted for publication once it meets all outstanding technical requirements.

Kind regards,

Antonio Medina Guerrero, Ph.D.

Academic Editor

PLOS ONE

Additional Editor Comments (optional):

Reviewers' comments:

Reviewer's Responses to Questions

**Comments to the Author**

1. If the authors have adequately addressed your comments raised in a previous round of review and you feel that this manuscript is now acceptable for publication, you may indicate that here to bypass the “Comments to the Author” section, enter your conflict of interest statement in the “Confidential to Editor” section, and submit your "Accept" recommendation.

Reviewer #1: All comments have been addressed

Reviewer #2: All comments have been addressed

2. Is the manuscript technically sound, and do the data support the conclusions?

Reviewer #1: Yes

Reviewer #2: Yes

3. Has the statistical analysis been performed appropriately and rigorously? 

Reviewer #1: Yes

Reviewer #2: Yes

4. Have the authors made all data underlying the findings in their manuscript fully available?

Reviewer #1: Yes

Reviewer #2: Yes

5. Is the manuscript presented in an intelligible fashion and written in standard English?

Reviewer #1: Yes

Reviewer #2: Yes

6. Review Comments to the Author

Reviewer #1: The authors have done a great job on the suggested changes, and have argued for the changes they did not see necessary.

For my part I think the manuscript is ready for publication, congratulations on a job well done and thanks for this manuscript on PBF microchemistry!

Reviewer #2: (No Response)

7. PLOS authors have the option to publish the peer review history of their article (what does this mean?). If published, this will include your full peer review and any attached files.

Reviewer #1: No

Reviewer #2: No

---

## [Editor Report · Acceptance letter]

4 Oct 2022

PONE-D-22-17432R1 

Otolith geochemistry reflects life histories of Pacific bluefin tuna 

Dear Dr. Mohan:

I'm pleased to inform you that your manuscript has been deemed suitable for publication in PLOS ONE. Congratulations! Your manuscript is now with our production department. 

Kind regards, 

on behalf of

Dr. Antonio Medina Guerrero 

Academic Editor

PLOS ONE